# Belonging and Inclusivity Make a Resilient Future for All: A Cross-Sectional Analysis of Post-Flood Social Capital in a Diverse Australian Rural Community

**DOI:** 10.3390/ijerph17207676

**Published:** 2020-10-21

**Authors:** Veronica Matthews, Jo Longman, James Bennett-Levy, Maddy Braddon, Megan Passey, Ross S. Bailie, Helen L. Berry

**Affiliations:** 1University Centre for Rural Health, The University of Sydney, Lismore, NSW 2480, Australia; jo.longman@sydney.edu.au (J.L.); james.bl@sydney.edu.au (J.B.-L.); maddy.braddon@sydney.edu.au (M.B.); megan.passey@sydney.edu.au (M.P.); ross.bailie@sydney.edu.au (R.S.B.); 2MAE Scholar, National Centre for Epidemiology and Population Health, Australian National University, Canberra, ACT 2600, Australia; 3Australian Institute of Health Innovation, Faculty of Medicine, Health and Human Sciences, Macquarie University, Sydney, NSW 2109, Australia; Helen@altitudeconsulting.com.au

**Keywords:** floods, mental health, social capital, inequality, Indigenous populations, low-income populations

## Abstract

In 2017, marginalised groups were disproportionately impacted by extensive flooding in a rural community in Northern New South Wales, Australia, with greater risk of home inundation, displacement and poor mental health. While social capital has been linked with good health and wellbeing, there has been limited investigation into its potential benefits in post-disaster contexts, particularly for marginalised groups. Six months post-flood, a cross-sectional survey was conducted to quantify associations between flood impact, individual social capital and psychological distress (including probable post-traumatic stress disorder). We adopted a community-academic partnership approach and purposive recruitment to increase participation from socio-economically marginalised groups (Aboriginal people and people in financial hardship). These groups reported lower levels of social capital (informal social connectedness, feelings of belonging, trust and optimism) compared to general community participants. Despite this, informal social connectedness and belonging were important factors for all participant groups, associated with reduced risk of psychological distress. In this flood-prone, rural community, there is a pressing need to build social capital collectively through co-designed strategies that simultaneously address the social, cultural and economic needs of marginalised groups. Multiple benefits will ensue for the whole community: reduced inequities; strengthened resilience; improved preparedness and lessened risk of long-term distress from disaster events.

## 1. Introduction

In disaster contexts, the value of close social networks is well documented for logistical, financial and emotional support, alleviating psychological stress following traumatic experiences [1]. Disaster management policies are increasingly drawing attention to investment in social resources as another form of ‘capital’ to help communities and individuals more effectively prepare, survive and recover from disaster events such as floods [2,3]. Social capital acts as ‘informal insurance’, facilitating a community’s collective action to accelerate recovery [4]. However, previous post-disaster research has shown that social capital does not always benefit everyone due to existing prejudices that may slow down recovery for marginalised groups [4].

Social capital has been variously described and measured either as individual perspectives or as community-level structures and characteristics [5]. Widely adopted in public health research, Putnam’s concept of social capital takes a macro-level approach, placing it as a collective resource strengthened by civic engagement, informal social connectedness, trust and social identity to facilitate group-level coordinated action with individual-level health consequences [5,6,7]. Putnam’s conceptualisation contains an implied causal mechanism whereby forms of community participation (e.g., volunteering) influence levels of social cohesion (e.g., social trust) [8,9].

Bonding, bridging and linking social capital describe network characteristics and flows of resources within and across groups: bonding refers to resources accessed within tightly knit groups of similar socio-economic and demographic profiles; bridging refers to resource flow between groups with weaker ties and different profiles, and; linking refers to resource flow across gradients of authority and power [7,10]. Where bonding social capital provides resources and support for ‘getting by’, bridging and linking social capital are important for ‘getting ahead’ [11]. All forms of social capital may work to promote health but they can also have costs and negative consequences for marginalised individuals [12,13], particularly where bonding capital reinforces exclusive social identities to the detriment of others external to the group [7,10]. Similarly, a lack of bridging capital reinforces social hierarchies [13]. Marginalised groups experience gaps in all forms of social capital [12,14,15] which may lead to increased health inequalities [10,16]. Therefore, having a better understanding of how social capital operates within a community may offer insights into how positive aspects (such as bridging ties) can be intentionally strengthened to more effectively address inequalities and improve the health and wellbeing of marginalised groups [5,13].

Social capital in health and resilience research is generally measured by its structural and cognitive components [6,8]. The structural component describes the nature and extent of community participation through which individuals develop social networks and the cognitive component describes the social cohesion resulting from community participation [8,9] or what people ‘do’ and ‘feel’ [17]. Personal social cohesion is assessed through individual subjective perceptions of levels of belonging, social trust (trust in strangers), generalised reciprocity (kindness of strangers) and optimism (hope for the future) [6]. Mental health may both be a product of or facilitator for social capital [9]. Longitudinal studies have demonstrated a positive, bi-directional relationship between mental health and structural components of social capital: better mental health leads to greater community participation/social connectedness and greater participation/connectedness leads to better mental health [9,18], including following a flooding event [19]. In this reciprocal relationship, social connectedness is a stronger, more consistent predictor of mental health than mental health is of social connectedness [18].

In 2017, record-breaking rainfall in Northern New South Wales (NSW) from ex-Tropical Cyclone Debbie (the second most destructive cyclone in Australia) caused widespread flooding, inundating local business districts and residential areas on a scale not seen in over forty years [20]. Shortly after, a community-academic partnership was formed to design and implement a study examining potential relationships between flood exposure and mental health and wellbeing outcomes [21]. Two Community Advisory Groups (CAGs) were established in Lismore and Murwillumbah, the main population centres of the region. They consisted of local health and community organisations, business groups and state and local authorities who have responsibility for flood planning, emergency response, mental health service provision and/or advocacy and support for particular subgroups within the community such as farmers, business owners, Aboriginal and Torres Strait Islander people and the socio-economically marginalised. Together with the CAGs, a conceptual framework was developed (the flood impact framework) which theorises pathways between flood exposure and psychological outcomes influenced by mediating factors at personal, community and organisational levels (e.g., socio-demographics, community cohesion, organisational disaster relief efforts) [21]. Based on published evidence, social capital was included as one of many potential mediators. It was predicted that greater levels of community participation and social cohesion would be protective against psychological distress and that this relationship would vary for different groups including marginalised people in the region. We define ‘marginalised’ as people with “… compromised or severely limited access to the resources and opportunities needed to fully participate in society and to live a decent life. Marginalised people experience a complex, mutually reinforcing mix of economic, social, health and early-life disadvantage, as well as stigma” (page 4 in [15]). A better understanding of how social processes work for these groups in a post-disaster context could improve the participatory co-design of resilience-building strategies, a process that in itself may promote social capital [22,23].

Northern NSW is a flood-prone region with over 30 flood disaster declarations in the decade from 2004 to 2014 [24]. Compared to state-level population characteristics, the Northern NSW rural region has higher proportions of people living with an underlying vulnerability, lower median household incomes and greater government income support reliance (e.g., single parent, disability, unemployment, and youth payments) [25]. The region also has a higher proportion of Australia’s First Nations people (4.1%) compared to the state average (2.9%) [26]. It is important to note that Aboriginal and Torres Strait Islander status does not in itself indicate marginalisation [15]; rather, it is the common intergenerational disadvantage and ongoing systemic racism that leads to a significant proportion experiencing marginalisation.

During the 2017 flood, marginalised groups (Aboriginal and Torres Strait Islander participants and participants in receipt of income support) were disproportionately impacted by the flood with a greater risk of home inundation, displacement and adverse mental health outcomes [27]. Despite substantial evidence that social capital can promote health and wellbeing, there has been limited empirical investigation into its potential mitigating effect against adverse psychological outcomes following weather-related disasters and how this may vary for marginalised groups. This study investigates at an individual level, associations between the components of social capital (community participation and personal social cohesion) and psychological distress following a major flood event in rural Australia. It examines how social capital has different effects on mental health for marginalised groups relative to other participants. Our aim is to use these findings to highlight what might or might not work in intervention design to assist community groups to strengthen social capital and adaptive capacity within this flood-prone region.

## 2. Materials and Methods

Data were taken from a cross-sectional survey of adults (16 years and older) in Northern NSW, six months after the region experienced extensive flooding. The questionnaire was formulated on the basis of the flood impact framework described above and outlined in our study protocol [21]. To minimise survey fatigue, the questionnaire contained instructions advising participants of the choice to complete a short version of the questionnaire (that included items on participants’ socio-demographic characteristics, flood exposure and their psychological health) or a longer version (all of the above as well as measures of community participation and personal social cohesion). A small prize draw (gift voucher for a local business) was offered as an incentive, with an increased number of entries given for completion of the full questionnaire. The prize draw was not advertised as part of the survey recruitment process.

To comprehensively understand the psychological impact within the community, we aimed to recruit participants from different socio-economic backgrounds experiencing different degrees of flood exposure. We utilised a local community-partnered purposive snowball sampling technique, where the CAGs reached out to their networked constituents offering support and encouraging completion of the questionnaire. This approach was particularly important for certain sectors of the community, as a degree of trust is required to engage socio-economically marginalised groups, including Aboriginal and Torres Strait Islander people and people living with disadvantage. For the purpose of this analysis, we defined the latter as recipients of the following types of income support as markers for chronic financial hardship and living with social marginalisation [15]: single parent support; unemployment support; youth allowance; disability support; and carer support. Our snowball sampling approach was supplemented by an extensive local media (print, broadcast and social media) and advertising campaign, including posters and paper surveys (with reply-paid postage) left in central community locations such as post offices, libraries, coffee shops and store-fronts of charitable organisations such as Lifeline, St Vincent de Paul and the Salvation Army. Project staff promoted the survey at various community events including farmers’ markets, and postcards were deposited in residential mailboxes with information on accessing the survey [21].

Our sampling approach resulted in a total of 2046 respondents completing the full version of the survey [21]. Given that most Aboriginal and Torres Strait Islander people in the Northern NSW area identify as Aboriginal, we respectfully use this term while recognising the diversity of First Nations culture that exists within the region. All participants gave their informed consent for inclusion before completing the questionnaire. The study was approved by the University of Sydney Human Research Ethics Committee (reference−2017/589) and the Aboriginal Health and Medical Research Council Human Research Ethics Committee (reference−1294/17).

### 2.1. Measures

Participants’ sociodemographic data included age, sex, Aboriginal and Torres Strait Islander status, relationship status, employment status, type of income support payments and educational qualifications. For flood exposure, a cumulative exposure index (CEI: range 0–5) was derived by summing the number of damage sites experienced out of five possibilities: suburb; non-liveable areas of their home (e.g., garden shed, garage); liveable areas of their home (e.g., bedrooms); income-producing property (business/farm); and the home of a significant other [21].

Self-report measures for post-flood distress included a single ongoing distress item from the Brief Weather Disaster Trauma Exposure and Impact Screen (‘Are you still currently distressed about what happened during the flood?’) [28] and the Post Traumatic Stress Disorder Checklist (PCL−6) [29], a brief clinical screening tool (cut-point for probable diagnosis ≥14) that was introduced as a list of ‘complaints’ that ‘people sometimes have’ after severe rain and flooding. Details of how the Brief Weather Disaster Trauma Exposure and Impact Screen was developed are presented in Appendix A; the measure was field-tested and deployed as part of the Queensland Government’s annual Self-Reported Health Status survey following severe flooding in the summer of 2010−11 [28]. It consists of four items adapted from previous research investigating post-traumatic stress disorder (PTSD) and depression following trauma in adults, adolescents and children within the Australian population. The yes/no ‘still currently distressed’ item from this measure was used for this analysis to allow for assessment of ongoing stress and anxiety related specifically to the flooding event (as distinct from anxiety arising from other causes) and for comparability to other similar studies in which it has been used [28]. For the PCL−6, respondents were asked to rate items on a 5-point Likert-type scale that evaluated experiences of intrusive memories, numbing/avoidance and hyper-arousal symptoms. The PCL−6 has shown adequate diagnostic performance in primary care settings including for minority populations (sensitivity 80–92%; specificity of 72–76%) [30,31]. Outcome variables were coded as binary for ongoing distress (yes/no) and probable PTSD (yes ≥ 14; no < 14).

The questionnaire included measures representing structural and cognitive constructs of social capital: community participation and personal social cohesion, respectively (Table 1). Previous research has proposed an association between these constructs with enhanced community participation building personal social cohesion which, in turn, positively influences mental health and wellbeing [6,8,9], including among Aboriginal respondents [32]. The extent of respondents’ agreement with statements that related to community participation and personal social cohesion was reported on a seven-point Likert-scale (the higher the score, the higher the level of agreement). We reversed the scoring for negatively worded statements. We utilised items from the Australian Community Participation Questionnaire that describe different domains of community participation: informal social connectedness (spontaneous, informal in-person connections); civic engagement (participation in organised activities) and political participation [33]. The use of social media was added as another form of community participation. The breadth of participation was measured by summing the number of participation activities (eleven in total, possible range 0–11). Individuals’ subjective perceptions of the quality and quantity of their community participation [6] were also measured. Personal social cohesion comprised an individual’s subjective perception of their sense of belonging (self-categorisation as belonging to a group and cognitive evaluation of the perceived social supports available for connecting, confiding and seeking help) [12,34], feelings of belonging (affective or emotional response to group membership) [6], social trust [12,35,36,37], generalised reciprocity [12,35]) and trait optimism [38]. Dispositional optimism (a tendency to expect good outcomes over bad) has been strongly linked to social trust and a sense of belonging and has been shown to be related to mental health within the Australian population [6,32]. For this reason, it is included as part of the concept of ‘personal social cohesion’, or the sense of social cohesion present in individuals.

Following data cleaning and coding, we examined the distribution of individual social capital items to determine appropriate analysis techniques. Where Likert-scale scores for the social capital measures were bimodal in distribution, we converted these to binary variables (scores 1–4 allocated 0: unsure or disagree; scores 4–7 allocated 1: agree). Since there was a mixture of ordinal and binary variables, polychoric correlations were used for subsequent confirmatory factor analysis (CFA) as outlined below.

### 2.2. Data Analysis

CFA was used to examine how well the previously defined measures of community participation and personal social cohesion fitted with our survey data [40]. For each of the social capital constructs described above, one-factor congeneric models were estimated on polychoric correlation matrices using maximum likelihood estimation with Stata software (StataCorp. 2017. *Stata Statistical Software: Release 15*. StataCorp LLC, College Station, TX, USA) and the user-written command -polychoric- (author Stas Kolenikov, 2016). To derive factor score weights for subsequent regression analysis, CFA was replicated in Amos (Arbuckle, J.L. (2006) Amos Version 25.0, Chicago: SPSS, USA) using asymptotically distribution-free estimation on raw data (polychoric correlation functionality unavailable), an appropriate technique for ordinal, non-normal data, small models and large sample sizes (>1000) [41]. Item loadings and fit statistics were comparable across the two estimation methods (Appendix B). Model goodness of fit was assessed using the comparative fit index (CFI—value of >0.95 indicates excellent model fit) and root mean square error of approximation (RMSEA—<0.05 indicates an excellent model fit, 0.05–0.08 indicates acceptable fit) [40]. Once optimal models were identified, we assessed internal consistency by calculating composite reliability scores using Jöreskog’s rho (acceptable score > 0.70).

Following identification of the one-factor congeneric models, two sets of composite measures were developed: unweighted (by taking the mean score of items within the composite); and weighted (taking mean score of items within the composite after applying factor score weights from the CFA). Descriptive statistics were produced for sociodemographic information and the unweighted social capital measures. Differences in sociodemographic variables and social capital scores across respondent groups (Aboriginal; financial hardship; and ‘other’ (or general respondent group)) were tested using independent sample t-tests/two proportions z-tests and Mann-Whitney U tests respectively. Kendall’s rank correlation coefficients (tau-b, T_b_) were calculated to examine the strength and direction of bivariate associations within respondent groups. Multiple hierarchical logistic regression models were tested to examine the independent contribution in prespecified order of items theorised to influence mental health outcomes following a flood (socio-demographic characteristics, flood exposure, community participation and social cohesion). While causality cannot be inferred from cross-sectional designs, hierarchical regression analysis allowed examination of the *plausibility* of the concept that community participation is associated with greater personal social cohesion which, together, supports positive mental health outcomes. Both weighted and unweighted social capital composite variables were tested in the models, however, there was no substantive difference between the analyses with respect to independent variables that significantly influenced mental health outcomes. Hence, unweighted results are reported as they are easier to interpret and replicate if needed in future analyses.

Prior to multivariate analysis, we tested for interactions between sociodemographic characteristics and (i) flood exposure and (ii) social capital variables to examine how the combination of personal factors with flood experience, social participation and social cohesion were associated with reporting each psychological issue. Given the number of interactions tested, we utilised a conservative *p*-value (<0.01), to guide the addition of statistically significant interactions to the relevant multivariate model step as described below.

Four blocks of variables (sociodemographic factors, flood exposure, community participation and personal social cohesion) were added sequentially to assess the unique proportion of variance each contributed to mental health problems. Tjur’s ‘coefficient of discrimination’ (*D*—the difference in mean of predicted probabilities of having symptoms of psychological distress versus no symptoms), analogous to the coefficient of determination (R^2^) in linear models, was used to evaluate the explanatory power of each block [42]. Non-significant contributors to explaining variance in psychological outcomes were removed from each step starting with the variable with the lowest standardised beta coefficient. Changes in beta values from one step to the next were examined to assess mediation effects in the relationship between community participation, social cohesion and mental health. The model was re-evaluated after each deletion until only significant predictors (*p*-value < 0.05) remained in each model. Odds ratios (ORs) and 95% confidence intervals (CIs) were reported for a total of six separate hierarchical logistic regression models calculated for two flood-related outcome measures (‘ongoing distress’ and ‘probable PTSD’) for each key interest group (Aboriginal respondents; respondents in receipt of financial hardship support; and ‘other’ respondents). Respondents who did not complete a health outcome measure were excluded from analysis for that indicator only.

## 3. Results

The CFAs were carried out on the full respondent dataset (*n* = 2046); results are detailed in Appendix B and summarised in Table 2. ‘Attending worship services’ (standardised loading = 0.22) was not strongly associated with the Civic Engagement construct. We included this item separately in subsequent regression analyses rather than attempt to fit it in a CFA. The WVS items measuring Generalised Reciprocity (‘most people try to be helpful’, ‘most people look out for themselves’) were weakly correlated in our dataset (polychoric ρ = 0.23). These, too, were added separately in regression analyses. The remaining items demonstrated acceptable scale reliability (ρ) and goodness of fit (CFI and RMSEA values) within their CFAs and were retained in one-factor model solutions (Table 2).

Of the total 2046 respondents who completed the full version of the survey, 1888 who provided complete sociodemographic data constituted the dataset for analysis. Of the respondent group, 3.5% (*n* = 67) were Aboriginal and 15% (*n* = 287) were respondents in financial hardship. Over one-third of Aboriginal respondents (*n* = 24) were also in receipt of types of income support related to chronic hardship. To obtain mutually exclusive groups and to minimise confounding, these were retained in the Aboriginal respondent group and excluded from the financial hardship category. Overall, the majority of respondents were women (69%, *n* = 1304) and aged between 45 to 64 years (53%, *n* = 995) (Table 3). Aboriginal and financially disadvantaged respondents were more likely to be younger, single, unemployed and have lower educational attainment. In the six months immediately following the flood, approximately one in five respondents was still distressed and one out of seven reported probable PTSD. There were higher proportions of Aboriginal and financial hardship respondents indicating ongoing distress and probable PTSD compared to ‘other’ respondents.

There were no significant differences in social capital scores between Aboriginal and hardship respondent groups (Table 4). However, informal social connectedness scores were significantly lower in both marginalised groups compared to ‘other’ respondents. Civic engagement and breadth of community participation (the number of different types of community activities participated in) was also significantly lower for respondents in financial hardship compared to ‘other’ respondents. For personal social cohesion, both marginalised groups had significantly lower levels of belonging, social trust and optimism compared to ‘other’ respondents.

In unadjusted analyses, Kendall rank correlation coefficients showed that higher severity of flood exposure was associated with higher levels of ongoing distress and probable PTSD at six months for all respondent groups (Table 5). As expected, most social capital variables were negatively correlated with psychological distress outcomes. Also, as predicted, community participation variables were less likely to be significantly associated with psychological distress compared to personal social cohesion variables (i.e., participation has a more distal influence on psychological outcomes compared to social cohesion). Informal social connectedness was significantly associated with ongoing distress only among ‘other’ respondents. Both informal social connectedness and civic engagement were associated with lower probable PTSD scores for respondents in receipt of financial hardship support and ‘other’ respondents. Among Aboriginal respondents only, higher social media engagement was associated with lower levels of ongoing distress and probable PTSD. Participating in a larger range of activities (greater breadth of participation) was significantly associated with lower probable PTSD scores for both financial hardship and ‘other’ respondents.

Higher levels of personal social cohesion were significantly associated with lower levels of probable PTSD in all respondent groups. Belonging and optimism were significantly associated with less ongoing distress for respondents in financial hardship. Similarly, these constructs, in addition to social trust, were associated with less ongoing distress for Aboriginal respondents (Table 5).

Table 6 and Table 7 summarise the unweighted hierarchical logistic regression results across all three respondent groups for ongoing distress and probable PTSD at six months respectively (weighted analyses produced trivial and non-significant differences in estimates with identical patterns of associations, so are not presented here). There were no significant interactions detected at *p* < 0.01 between sociodemographic characteristics and flood exposure or social capital variables.

### 3.1. Aboriginal Respondents

None of the socio-demographic factors for Aboriginal respondents made an independent contribution to explaining their ‘still distressed’ status six months after the flood. Higher levels of flood exposure were strongly associated with ongoing distress (aOR 2.73; 95% CIs: 1.52–4.91) and remained that way in the final model, explaining most model variance (change in Tjur’s D = 22%) (Table 6). After adjusting for socio-demographic characteristics and flood exposure, social media engagement was not significantly associated with ongoing distress. While types of community participation were not significant in the model, enjoyment of participation was strongly associated with less distress (aOR 0.59; 95% CIs: 0.37–0.95). None of the personal social cohesion variables was independently significantly associated with ongoing distress for this respondent group.

Compared to ongoing distress, there were different patterns of association between flood exposure, social capital and probable PTSD for Aboriginal respondents (Table 7). Higher levels of educational attainment made a significant independent contribution to explaining lower probable PTSD scores. This variable became non-significant when flood exposure was added to the model. Flood exposure was associated with a higher risk of probable PTSD explaining a further 7% of the model. Greater informal social connectedness was significantly independently associated with lower PTSD risk, while perceptions about the quality and quantity of time spent with others did not further explain PTSD outcomes. The contribution of flood exposure and informal connectedness became non-significant with the addition of the social cohesion variables. Feelings of belonging (aOR 0.41; 95% CIs: 0.23–0.71) were strongly associated with lower levels of probable PTSD and explained most of the model variance (18%) for Aboriginal respondents.

In summary, in the final models, consistent with predictions in our flood impact framework, post-flood ongoing distress was explained in order of magnitude by greater levels of flood damage and lower scores of enjoying social participation. A greater risk of post-flood probable PTSD was mainly explained by lower feeling of belonging scores.

### 3.2. Respondents in Financial Hardship

Socio-demographic variables were not significantly associated with ongoing distress for respondents in financial hardship six months after the flood. Similar to Aboriginal respondents, higher levels of flood exposure were strongly associated with ongoing distress (aOR 1.86; 95% CIs: 1.46–2.38) explaining most of the model variance (10%) (Table 6). Neither type nor perceptions of community participation made any contribution to explaining ongoing distress. Greater optimism (aOR 0.62; 95% CIs: 0.48–0.79) was the only component of social cohesion that was significantly associated with lower levels of ongoing distress, explaining a further 5% of the variance in the model.

Similar to ongoing distress patterns of association, socio-demographic factors were not significantly associated with probable PTSD and greater flood exposure was strongly associated with a higher risk of probable PTSD (1.63; 95%CIs: 1.30–2.05) explaining 7% of the model variance (Table 7). In contrast to ongoing distress, informal social connectedness (aOR 0.71; 95% CIs: 0.56–0.89), enjoying participation (aOR 0.76; 95%CIs: 0.61–0.95) and having sufficient quantity of social time (aOR 1.30; 95% CIs: 1.08–1.56) were significantly associated with probable PTSD. Increased feelings of belonging (aOR 0.48; 95% CIs: 0.37–0.62) was the only social cohesion variable that was significantly associated with lower probable PTSD scores. The addition of feelings of belonging explained a further 9% of the variance and rendered the community participation indicators non-significant in the probable PTSD model.

As predicted, in the final models for respondents in financial hardship, post-flood distress was explained in order of magnitude by greater levels of flood exposure and lower optimism scores. Post-flood probable PTSD was explained in order of magnitude by greater flood exposure and lower feeling of belonging scores.

### 3.3. General Community Respondents

Socio-demographic variables for ‘other’ respondents were not significantly associated with ongoing distress six months after the flood (Table 6). As with both marginalised respondent groups, higher levels of flood exposure were strongly associated with reports of ongoing distress (aOR 2.15; 95% CIs: 1.90–2.42) explaining most variance in the model (13%). Unlike marginalised respondent groups, there was a significant association between higher levels of informal social connectedness and less distress (aOR 0.86; 95%CIs: 0.77–0.97). Similar to Aboriginal respondents, enjoying community participation was significantly associated with less ongoing distress for the general respondent group (aOR 0.76; 95% CIs: 0.67–0.87). Having a greater sense of belonging (perceived social supports) (aOR 0.81; 95% CIS: 0.68–0.96) and optimism (aOR 0.74; 95% CIs: 0.64–0.86) were also significantly associated with less distress. The contribution made by informal connectedness and enjoying community participation became non-significant when these social cohesion variables were added to the model.

Lower educational attainment, not being in paid employment and single relationship status made independent contributions to increasing the risk of probable PTSD for the general respondent group (Table 7). These demographic factors, however, became non-significant in subsequent model steps. Again, like both marginalised respondent groups, higher levels of flood exposure were strongly associated with probable PTSD (aOR 2.22; 95% CIs: 1.91–2.58). Unlike marginalised groups, however, flood exposure explained most variance in probable PTSD outcomes for general community respondents (10%). There were similar patterns of association between social capital variables and probable PTSD between the general respondent group and those in financial hardship. Higher informal social connectedness (aOR 0.72; 95% CIs: 0.63–0.83) and enjoying social participation (aOR 0.60; 95% CIs: 0.51–0.70) were significantly associated with lower probable PTSD scores. Wanting to spend more time with others (indicating a degree of social isolation; aOR 1.16; 95%CIs: 1.02–1.32) was significantly associated with an increased risk of probable PTSD. Of all community participation variables, only enjoyment of participation remained significant in the final model for ‘other’ respondents. Like marginalised groups, lower scores for feelings of belonging (aOR 0.65; 95% CI: 0.55–0.76) were associated with higher probable PTSD scores. In addition, however, greater optimism (aOR 0.67; 95% CI: 0.55–0.81) was also strongly associated with less PTSD symptomology for the general respondent group.

In summary, significant associations in the final models align with predictions in our *flood impact framework*. Post-flood distress was explained in order of magnitude by greater flood exposure and lower optimism and a sense of belonging scores (perceived availability of social supports). Post-flood probable PTSD was explained by greater flood exposure and lower quality of social participation, feelings of belonging and optimism scores.

## 4. Discussion

Broadly, our findings support the propositions that (i) the components of social capital may be causally related in that community participation may be an important contributor to the formation of social cohesion; and (ii) while exposure to a flood event harms mental health across the whole community, the mental health of those with more social capital is not as severely harmed as those with less social capital. We examined the relationship between social capital and mental health among Aboriginal, financially disadvantaged and other members of the general community six months following a severe flood event. As expected, the greater participants’ exposure to the flood, the greater the likely harm to their mental health, particularly so for marginalised community members. Social capital played an important role in the degree of flood-related harm people reported in that those with higher levels of social capital reported less harm to their mental health than did those with less. However, the strength and nature of this effect varied by the group.

### 4.1. Aboriginal Respondents

With lower levels of informal social connectedness, belonging, social trust and optimism, Aboriginal respondents had less social capital than the general respondent group. These findings are in line with other social capital analyses in Aboriginal population-representative surveys [32,43,44]. As in previous studies, we found subtle differences in what mattered most for mental health and wellbeing compared to other respondent groups. Aboriginal respondents were like other groups in that individuals with greater feelings of belonging were less likely to experience post-flood PTSD. In contrast to other groups, optimism did not feature amongst the social cohesion factors that mattered most for Aboriginal respondents in terms of reducing the likelihood of ongoing distress.

Social capital and resilience can mean different things for different populations, suggesting that the way it is measured in the general Australian population may not adequately capture concepts of social participation and cohesion important to Aboriginal communities [32,45]. The community participation variables used in this study have been validated previously in an Australian Aboriginal community [6,30] and our study confirms the relevance of the participation variables (including social media engagement as a new type of participation) to Aboriginal participants. Yet, from an Aboriginal perspective, there are other characteristics of social relationships and resilience that are important in overcoming adversity. Relational identity is key, that is, the knowledge of and connection to one’s own community, culture and Country [46]. Colonisation severely disrupted these connections, the impact of which is still acutely felt today. Land dispossession, social and cultural dislocation (including the destruction of languages) and systematic genocide (including the forced removal of children from their Aboriginal families) have led to inter-generational trauma with devastating consequences for social and emotional wellbeing. Systemic and interpersonal racism reinforces socio-economic exclusion and mistrust in mainstream institutions [44,45] and has been linked to depression in Aboriginal people [47]. Consequently, there are significant chronic disparities across socio-economic and health indicators between Aboriginal and non-Indigenous Australians. The active resistance by and survival of Aboriginal communities throughout history and against ongoing adversity speaks to their strength, resilience and determination. The cultural context of this resilience (strong familial links, connection to country, language and ceremony) is protective in the face of repeated tragedies that Aboriginal communities often experience [48,49] and our study provides further evidence of how this may operate in the face of natural disasters.

While a strong sense of shared identity and belonging (bonding capital) within Aboriginal communities is important for their resilience and wellbeing, there is complexity in the link between Aboriginal social capital and social mobility. In the general community, connecting to other groups with different social identities has the potential to help one ‘get ahead’ by making accessible new opportunities and resources [11]. To receive some form of mutual benefit in this way intrinsically involves trust and reciprocity with an expectation of some form of ‘repayment’ (the amount and timing of which is not fixed) [50]. Considering the historical and cultural contexts described above, the pursuit of broader linkages (bridging capital) for Aboriginal people may be limited where their trust in members of the general community is compromised and their within-community social capital may not be valued or have currency outside of their community due to racial prejudices [45].

Despite the importance of historical and cultural contexts, consideration of these contexts is not currently evident in the development of local-level disaster risk reduction strategies. Active and equal participation of and leadership by Aboriginal people has resulted in successful public health responses to entrenched domestic violence within a community [48] and in prioritising the safety of Aboriginal communities during the current COVID−19 pandemic [51], demonstrating the importance and effectiveness of culturally-led solutions to complex threats to health and wellbeing. In a similar way, there is a great opportunity for Aboriginal-led approaches to address disaster risk that would benefit the whole community. For instance, Caring for Country initiatives, where Aboriginal and Torres Strait Islander knowledge is used appropriately to care for traditional lands and seas, have continually demonstrated multiple social, cultural, ecological, economic and health benefits [52,53,54]. These Aboriginal-led partnerships strengthen culture as well as enhance respect and appreciation of Aboriginal knowledge within mainstream populations [54]. By focusing on cultural context, strengthening connection to Country and increasing social networks, such initiatives will likely enhance feelings of belonging for Aboriginal people, a key driving factor influencing post-disaster distress.

A novel finding from this study is that social media may be a promising avenue for strengthening informal social connectedness for Aboriginal communities. Compared to the general community and those in financial hardship, Aboriginal respondents with higher social media usage were less likely to indicate post-flood distress and PTSD, perhaps because it increases social connectedness in this group. Previous research has shown social media use to be more common among Aboriginal compared to non-Indigenous people [55]. There is complexity in the relationship between the use of technology and social connectedness. Whether it enhances the quality of social relationships depends on the type of platform, motives for use and whether it is used actively or passively which, in turn, are influenced by socio-demographic characteristics [56,57]. In this study, the relationship between social media and distress for Aboriginal respondents was non-significant after controlling for socio-demographic characteristics, indicating that these characteristics may mediate the relationships. A more nuanced understanding is required to develop strategies to enhance its effectiveness in reducing isolation for this group. Social media can be an effective tool if used to strengthen existing relationships or initiate new meaningful ones (rather than as a substitute for real-life interaction) [57]. It may also be an effective vehicle for managing disaster risk and providing health messaging and education [55,58].

### 4.2. Respondents Living with Financial Disadvantage

Like Aboriginal community members, people living with financial disadvantage (as indicated in this study by being in receipt of certain types of government income support), had less social capital than general community members (including lower levels of informal social connectedness, civic engagement, belonging, social trust and optimism) supporting other research showing income inequality to be a consistent predictor of community participation [59], social isolation and sense of belonging [60].

Compared to general community members, those in financial hardship were more likely to be single, unemployed and have lower educational attainment levels. Quality of time spent socially and feelings of belonging were what mattered most for those in financial hardship with respect to probable PTSD outcomes. As a corollary, those wanting to increase the quantity of time spent socially (social isolation) were more likely to experience post-flood PTSD. Reasons for social isolation can be structural (i.e., lack of resources to enable access to social activities; lack of opportunity to access social networks otherwise available through education or employment); interpersonal (i.e., being avoided by others due to prejudice and discrimination); and personal (e.g., embarrassment, concern about stigmatisation or poor health) [60]. Because of these issues, people in financial hardship generally avoid social situations perceived as challenging, tending instead to socialise with others experiencing the same marginalisation. As a result, they generally have commensurately smaller and less reciprocal networks [60,61]. Places of belonging for the financially marginalised tend to be community support agencies or drop-in centres due to the economic and social support they provide. While relationships generated with service providers (e.g., providing food, housing, employment support, etc.) are beneficial, they are not spontaneous relationships but are ‘deliberately constructed’ and do not necessarily meet the social needs of marginalised people [61]. Similar to Aboriginal people, bonding social capital is an important buffer against poor mental health while lack of bridging social capital can be detrimental. For example, low-income individuals living in affluent areas can have worse mental health (exacerbated by social exclusion) compared to those living in deprived neighbourhoods [14,59].

People in financial hardship with greater optimism (a tendency to expect positive outcomes in the future), were less likely to experience ongoing distress. Optimists refuse to give up [62]. Instead, they tend to look for benefits in adversity and employ more effective coping strategies than pessimists, making them more resilient to stressful events [63]. This is relevant to coping with a flood: optimism moderates the relationship between the level of household damage in a disaster and personal recovery [64]. Optimists’ persistence in overcoming personal obstacles has also been attributed to their ability to forge bridging relationships across demographic and socio-economic divides [63]. In this study, greater informal social connectedness was related to greater optimism for people in financial hardship and associated with lower levels of ongoing distress. Resilience-building strategies for financially marginalised groups may benefit from interventions that build meaningful bridging relationships in environments that are safe and enjoyable from their perspective [6]. Such co-designed initiatives, preferably simultaneously addressing economic needs, will enhance agency and hope for the future [65].

### 4.3. Other Members of the General Community

Less optimistic members of the general community were more likely to show signs of post-flood distress and PTSD. This concurs with previous post-disaster research showing optimism reduces the likelihood of developing PTSD, suggesting a possible pathway to improve recovery and prevent adverse mental health impact [64]. General community members with a sense of belonging were also less likely to indicate long-term distress. It makes intuitive sense that post-disaster distress can be mitigated for individuals by turning to emotional, financial and social supports available through personal networks for recovery assistance. As for marginalised groups, greater feelings of belonging (the emotional evaluation of connectedness) decreased the likelihood of post-flood PTSD. Belonging is a fundamental human need [66]. There is a critical link between belonging and shared social identity and a belief that one’s life is meaningful which is important for wellbeing across different social groups, particularly for those that experience systematic social exclusion [60,66].

### 4.4. Belonging and Inclusivity Make for a Resilient Future

Feelings of belonging that are enhanced, possibly created, by participation and social inclusion are key to alleviating post-flood distress for this diverse rural community. Belonging and shared identity are multifaceted, comprising our material possessions, immediate and extended social networks as well as the place we call home [67]. Receiving increasing attention in post-disaster recovery research is the psychology of place (incorporating social and geographical contexts) and the concept of ‘solastalgia’ [67,68,69]. In NSW rural communities, feelings of belonging and perceptions of one’s environment are important for resilience [70]. Perhaps reflecting Aboriginal notions of connection to Country and its importance for wellbeing, solastalgia describes the sense of loss experienced by individuals when the surrounding environment changes to the extent that it no longer resembles home or becomes a place of danger in a disaster-prone area [68]. Extreme events that destroy homes and livelihoods or which force evacuation and long periods of displacement are known to exacerbate mental health issues, particularly for marginalised groups [27,67].

Given the complexity of social capital and the subtle variation in how it operates across different socio-economic groups, approaches to developing resilience strategies must involve the very groups for which they are designed. This analysis has pointed out key issues that may work to boost social connectedness for marginalised groups. In-depth qualitative research is required to fully understand the contextual and cultural factors that shape the specific needs of these different groups to jointly enhance participation and social cohesion for improved community adaptive capacity and disaster resilience. Compared to urban areas, rural communities tend to be known for high levels of some social capital (such as community participation and trust) but they can also have lower levels of tolerance for diversity, undermining their ‘collective efficacy’ [71]. So, while participatory approaches are critical, it is important that intervention strategies not be compartmentalised within social groups. Rather, we need to design strategies that consider broader contexts and are structured to be inclusive (e.g., interactions between social groups) to maximise the effectiveness of social capital interventions to strengthen overall community resilience.

### 4.5. Strengths and Limitations

Our sampling approach, while necessary to meet the goals of this study, constrains our ability to generalise our results to the broader population. Further, this is a self-report, cross-sectional design that limits our ability to untangle complex pathways to determine cause and effect and the presence of bi-directional relationships between social capital and mental health. Hence, our study design does not permit conclusions about whether social capital was directly protective against flood-related harm to mental health. Pre-existing mental health status may have biased responses and without pre-disaster community participation and social cohesion measures, we cannot be sure how the flood influenced social capital across the respondent groups.

While the proportion of Aboriginal respondents was close to the proportion living within Northern NSW, the small number of Aboriginal respondents reduced statistical power and may have led to the exclusion of meaningful predictors of flood-related distress. Where sample numbers were small, our analysis focused largely on the direction of associations and whether they were consistent with our expectations of the relationships between social capital, flood exposure and psychological distress. Our results were consistent with other studies investigating Aboriginal and Torres Strait Islander social capital [32,43,44] and can usefully inform future research with this population in the co-design of disaster risk reduction strategies. While validation studies of the Australian Community Participation Questionnaire and feelings of belonging included an Aboriginal community [6,32], our other social capital measures have been wholly designed and validated within so-called Western populations and may not adequately represent the experiences of other cultural groups. We also recognise that social capital for groups cannot be understood in isolation, but as part of an interacting set of capitals within the community that encapsulates human (knowledge, skills, the health of individuals), natural (land, water and biological resources), physical (infrastructure, equipment and technological resources) and financial (income, savings, credit, etc.) dimensions that also influence the adaptive capacity of rural communities [72].

Despite these limitations, our findings are consistent with our expectations and with other studies that have used population-representative samples and other study designs. We aimed to use a theoretically-driven approach to describe and quantify the relationships between flood impact, social capital and mental health with a particular focus on comparing the experiences of different types of community members. Using directly flood-related measures of mental health and adjusting for a very wide range of relevant socio-demographic controls, we found support for our proposition that social interactions, supports and cohesion are important in mitigating distress related to the flood.

A particular strength of our study was the close engagement with the community which led to our pragmatic, purposeful sampling approach that enabled measurement of these theoretical relationships for diverse, vulnerable sub-population groups. The CAGs continued to meet regularly over a period of 18 months during which findings were shared and interpretative discussions held to inform report writing and the dissemination of findings [21]. The aim of the community-academic partnership was to undertake useful research and disseminate findings addressing community-driven information needs. Our theories were supported by the findings which provide new insights on the development of local public health and disaster management policies aimed at strengthening dimensions of social capital to reduce post-disaster mental health. With Northern NSW being a flood-prone area [24], it is inevitable that this region will experience similar disasters in the future. There is a pressing need therefore to strengthen community social capital collectively through co-designed strategies that simultaneously address social and economic exclusion, cultural needs and environmental restoration. Multiple benefits for the community will ensue: reduced inequities; strengthened psychological well-being and resilience; lessened risk of long-term personal distress from disaster events; and reduced need for expensive individual psychological interventions [73] which are inequitably available and accessed [74,75].

## 5. Conclusions

Following the 2017 Northern NSW flood, Aboriginal and financially disadvantaged respondents reported lower levels of social capital (informal social connectedness, feelings of belonging, trust and optimism) compared to general community participants. Despite this, informal social connectedness and belonging were important factors for all participant groups and were associated with reduced risk of ongoing distress and PTSD.

Although it is well established that social capital is vital to promoting and maintaining positive mental health and wellbeing, there is relatively little research on how social capital influences psychological outcomes from weather-related disasters and, specifically, for marginalised population groups. Our study has deconstructed social capital to highlight what matters most for socio-economically marginalised groups to inform tailoring of safe and effective resilience-building strategies. Access to social capital is not homogeneous, with various groups subject to differential barriers in building and benefitting from social capital and its benefits to mental wellbeing. Community-level interventions are required tailored to specific groups through participatory processes. Future studies will be able to further disentangle these concepts, especially with regard to cause and effect, and to study how social capital operates in broader community contexts: which social resources benefit health for individual groups; and which characteristics of the wider social environment may promote such benefits.

## Figures and Tables

**Table 1 ijerph-17-07676-t001:** Social capital measures used within the Northern NSW Community Recovery after Flood survey.

Construct	Items	Source
	Community Participation	
Informal Social Connectedness	I make time to keep in touch with my friends; I chat with my neighbours when I see them; I spend time with extended family members (relatives who don’t live with me)	Australian Community Participation Questionnaire (ACPQ) [33]
Social Media Engagement	I am active on social media (e.g., Facebook, Snapchat, Instagram)	New
Civic Engagement	I take part in community-based clubs or associations (e.g., Rotary, CWA, book club, Lions); I go to arts or cultural events; I attend community events such as farmers’ markets, festivals and shows; I take part in sports activities or groups; I volunteer locally (e.g., Meals on Wheels, school fete, Rural Fire Service); I attend worship services or go to prayer meetings	ACPQ [33]
Political Participation	I get involved with political activities (e.g., through interest groups, public meetings, rallies)	Adapted from ACPQ [33]
Perceptions of Participation	I enjoy the time I spend with others socially; I would like to spend more time with others socially	Adapted from Berry, 2008 [39]
**Construct**	**Personal Social Cohesion**	**Source**
Sense of Belonging	When I feel lonely, there are several people I could call and talk to; I have family or friends I can confide in; I feel that I’m on the fringe in my circle of friends; I don’t often get invited to do things with others; There are people outside my household who can offer help in a crisis.	Adapted from Interpersonal Support Evaluation List (ISEL) [34]
Feelings of Belonging	I feel like an outsider; I feel that I belong; I feel included.	Adapted from Berry (unpublished)
Social Trust	Most people keep their word; Most people do what they say they’ll do; Most people around here succeed by stepping on others; Most people tell the truth when they’re sorting out a problem; You can’t be too careful with some people; Most people can be trusted.	Adapted by Berry & Rodgers [36] from Organisational Trust Inventory (OTI) [37] & World Values Survey (WVS) [35]
Generalised Reciprocity	Most people try to be helpful; Most people look out for themselves	Adapted from WVS [35]
Trait Optimism	Overall, I expect more good things to happen to me than bad; In uncertain times, I always expect the best; If something can go wrong for me, it will; I’m always optimistic about my future	Selected from Life Orientation Test – Revised [38]

**Table 2 ijerph-17-07676-t002:** Confirmatory Factor Analysis for composite social capital constructs using polychoric correlation matrices (*n* = 2046).

Construct	Factor Loadings (Range)	CFI	RMSEA	95%CI	ρ Reliability
Informal Social Connectedness	0.60–0.83	1.000	0.000	(0.000–0.040)	0.72
Civic Engagement	0.45–0.81	0.991	0.058	(0.041–0.078)	0.73
Sense of Belonging	0.43–0.86	0.997	0.048	(0.028–0.071)	0.75
Feelings of belonging	0.67–0.88	1.000	0.000	(0.000–0.050)	0.85
Social Trust	0.36–0.82	0.997	0.032	(0.016–0.049)	0.77
Trait Optimism	0.55–0.88	1.000	0.029	(0.000–0.073)	0.82

CFI: Comparative Fit Index; RMSEA: root mean square error of approximation; 95% CI: Confidence Interval.

**Table 3 ijerph-17-07676-t003:** Demographic profile and mental health outcomes by respondent group.

Characteristic	Category	Aboriginal Respondents (*n* = 67; 3.5%)		Respondents in Financial Hardship (*n* = 287; 15.2%)		Other Respondents (*n* = 1534; 81.3%)		Total (*n* = 1888)	
		Mean	SD	Mean	SD	Mean	SD	Mean	SD
Age		46.5 ^##^	14.0	48.8 ^###^	13.0	52.4	14.4	51.7	14.3
		***n***	**%**	***n***	**%**	***n***	**%**	***n***	**%**
Sex	Female	49	73.1	197	68.6	1058	69.0	1304	69.1
Employment	Not in employment ^	15	22.4 ***	132	46.0 ***	144	9.4	291	15.4
Education	University level	20	29.9 ^##^	88	30.7 ^###^	735	47.9	843	44.7
Relationship status	Single	31	46.3 ***	178	62.0 ***	401	26.1	610	32.3
Mental health outcomes	Ongoing distress	28	41.8 ***	92	32.1 ***	305	19.9	425	22.5
	Probable PTSD	24	35.8 ***	94	32.8 ***	173	11.3	291	15.4

^ In addition to respondents looking for paid work or unable to work due to long-term illness, ‘not in employment’ also includes respondents of working age in full-time education, looking after family and home and/or doing regular unpaid volunteer work. Mean/proportion of respondents within the marginalised group is significantly greater (*) or smaller (#) than the mean/proportion in ‘other’ respondents *^,#^
*p* < 0.05; **^,##^
*p* < 0.01; ***^,###^
*p* < 0.01.

**Table 4 ijerph-17-07676-t004:** Medians and interquartile ranges (IQR) for social capital variables in three respondent groups (higher scores indicate greater agreement with perception statements; *n* = 1888).

Social Capital Construct	Aboriginal Respondents (*n* = 67)			Financial Hardship Respondents (*n* = 287)			Other Respondents (*n* = 1534)	
	Med.	IQR		Med.	IQR		Med.	IQR
Community participation (score range 1–7)								
Informal Social Connectedness	5.3	(4.0–6.0)	**	5.0	(4.0–6.0)	***	5.7	(4.7–6.0)
Social Media Engagement	5.0	(4.0–6.0)		5.0	(4.0–6.0)		5.0	(3.0–6.0)
Civic Engagement	4.0	(2.8–5.0)		4.0	(3.0–4.8)	***	4.2	(3.2–5.2)
Religious Engagement	2.0	(1.0–4.0)		1.0	(1.0–3.0)	*	2.0	(1.0–4.0)
Political Participation	4.0	(1.0–5.0)		4.0	(2.0–5.0)		3.0	(2.0–5.0)
Breadth of participation (0–11)	6.0	(4.0–7.0)		5.0	(3.0–7.0)	***	6.0	(4.0–8.0)
Perceptions of participation (1–7)								
Enjoyment (enjoy the time spent socially)	6.0	(5.0–6.0)	**	6.0	(5.0–6.0)	***	6.0	(5.0–7.0)
Sufficiency (desire to spend more time socially)	5.0	(4.0–6.0)		5.0	(4.0–6.0)		5.0	(4.0–6.0)
Personal Social Cohesion (1–7)								
Sense of Belonging	4.8	(4.0–6.0)	**	4.8	(4.0–5.6)	***	5.4	(4.6–6.0)
Feelings of Belonging	5.0	(3.3–6.0)	*	4.3	(3.3–5.7)	***	5.3	(4.3–6.0)
Social Trust	4.2	(3.3–4.8)	***	4.0	(3.5–4.7)	***	4.7	(4.0–5.2)
Reciprocity—People try to help	5.0	(4.0–6.0)		5.0	(5.0–6.0)		5.0	(5.0–6.0)
Reciprocity—People look after themselves	5.0	(4.0–6.0)		5.0	(4.0–6.0)		5.0	(4.0–6.0)
Optimism	4.5	(3.5–5.8)	***	4.5	(3.8–5.3)	***	5.3	(4.3–5.8)

** p* < 0.05; ** *p* < 0.01; *** *p* < 0.001: Mann-Whitney U tests compare mean rank of scores between Aboriginal and ‘other’ respondents and financial hardship respondents and ‘other’ respondents. (Note: Two distributions may have equivalent medians but different rank sums. For example, enjoyment of community participation scores, marginalised respondent groups had lower rank sums (other than those at the median) compared to ‘other’ respondents.).

**Table 5 ijerph-17-07676-t005:** Kendall Rank Correlation Coefficients between social capital variables and mental health outcomes for each respondent group.

Social Capital Construct	Aboriginal Respondents (*n* = 67)				Financial Hardship Respondents (*n* = 287)				Other Respondents (*n* = 1534)			
	Ongoing Distress		PTSD		Ongoing Distress		PTSD		Ongoing Distress		PTSD	
Flood Exposure ^#^	0.39	***	0.22	*	0.29	***	0.24	***	0.31	***	0.26	***
Community Participation												
Informal Social Connectedness	−0.04		−0.13		−0.01		−0.15	**	−0.06	*	−0.09	***
Civic Engagement	−0.04		−0.10		−0.001		−0.11	*	−0.03		−0.07	**
Social Media Engagement	−0.25	*	−0.25	*	−0.03		−0.06		0.01		−0.01	
Religious Engagement	0.04		−0.10		−0.03		−0.08		0.001		−0.04	
Political Participation	0.06		−0.01		0.04		−0.02		0.03		−0.01	
Breadth of Participation	−0.03		−0.18		−0.04		−0.11	*	−0.03		−0.09	***
Perceptions of Participation												
Enjoyment of time socialising	−0.24	*	−0.23	*	−0.08		−0.17	**	−0.14	***	−0.20	***
Sufficiency of time socialising	0.02		0.04		0.09		0.07		−0.01		0.01	
Personal Social Cohesion												
Sense of Belonging	−0.23	*	−0.38	***	−0.12	*	−0.29	***	−0.14	***	−0.17	***
Feeling of Belonging	−0.29	**	−0.42	***	−0.15	**	−0.35	***	−0.13	***	−0.21	***
Social Trust	−0.23	*	−0.34	**	−0.08		−0.18	**	−0.11	***	−0.14	***
Reciprocity—people try to help	−0.22		−0.39	***	−0.03		−0.17	**	−0.09	***	−0.11	***
Reciprocity—people look after themselves	0.18		0.27	*	0.03		0.004		0.05	*	0.08	***
Optimism	−0.21	*	−0.24	*	−0.19	***	−0.24	***	−0.16	***	−0.20	***

** p* < 0.05; ** *p* < 0.01; *** *p* < 0.001; ^#^ Cumulative Exposure Index (CEI).

**Table 6 ijerph-17-07676-t006:** Parameter estimates and associated statistics of multiple hierarchical logistic models predicting flood-related ongoing distress for each respondent group, controlling for sociodemographic factors ^‡^.

	Aboriginal Respondents (*n* = 66)				Financial Hardship Respondents ^†^ (*n* = 280)				Other Respondents (*n* = 1477)			
Model Block	aOR	(95%CI)	∆*D*	*D*	aOR	(95%CI)	∆*D*	*D*	aOR	(95%CI)	∆*D*	*D*
1. Flood Exposure (CEI)			0.22	0.29			0.10	0.11			0.13	0.14
	2.73	(1.52–4.91)	**^		1.86	(1.46–2.38)	*** ^		2.15	(1.90–2.42)	*** ^	
2. Community Participation												
2 A. Type & extent of participation											0.01	0.15
Informal Social Connectedness	-				-				0.86	(0.77–0.97)	*	
2 B. Perceptions of participation			0.05	0.34							0.01	0.16
Enjoy time spent socially	0.59	(0.37–0.95)	* ^		-				0.76	(0.67–0.87)	***	
3. Personal Social Cohesion							0.05	0.16			0.02	0.18
Sense of Belonging	-				-				0.81	(0.68–0.96)	* ^	
Optimism	-				0.62	(0.48–0.79)	*** ^		0.74	(0.64–0.86)	***^	

^‡^ Age, sex, education level, employment and relationship status; ^†^ In receipt of following income support: single parent payment, unemployment allowance, youth allowance, disability support, carer payment; D = Tjur’s coefficient of discrimination; * *p* < 0.05; ** *p* < 0.01; *** *p* < 0.001; ^ Predictor made an independent significant contribution in the third and final model; adjusted odds ratios (aORs) reported are for the model in which the predictors were added.

**Table 7 ijerph-17-07676-t007:** Parameter estimates and associated statistics of multiple hierarchical logistic models predicting flood-related probable PTSD for each respondent group, controlling for sociodemographic factors ^‡^.

	Aboriginal Respondents (*n* = 67)				Financial Hardship Respondents ^†^ (*n* = 283)				Other Respondents (*n* = 1463)			
Model Block	aOR	(95%CI)	∆*D*	*D*	aOR	(95%CI)	∆*D*	*D*	aOR	(95%CI)	∆*D*	*D*
Socio-demographic Factors				0.12								0.02
Education (non-university level)	4.56	(1.12–18.60) *			-				1.68	(1.20–2.35) **		
Employment (not in employment)	-				-				2.08	(1.31–3.29) **		
Relationship status (single)	-				-				1.44	(1.02–2.05) *		
1. Flood Exposure (CEI)			0.07	0.19			0.07	0.09			0.10	0.12
	1.69	(1.06–2.72) *			1.63	(1.30–2.05) ***^			2.22	(1.91–2.58) ***^		
2. Community Participation												
2 A. Type and extent of participation			0.08	0.27			0.03	0.12			0.02	0.14
Informal Social Connectedness	0.53	(0.31–0.92) *			0.71	(0.56–0.89) **			0.72	(0.63–0.83) ***		
2 B. Perceptions of participation							0.04	0.16			0.04	0.18
Enjoy time spent socially	-				0.76	(0.61–0.95) *			0.60	(0.51–0.70) ***^		
Sufficient time socialising	-				1.30	(1.08–1.56) **			1.16	(1.02–1.32) *		
3. Personal Social Cohesion			0.18	0.45			0.09	0.25			0.06	0.24
Feeling of Belonging	0.41	(0.23–0.71) ** ^			0.48	(0.37–0.62) *** ^			0.65	(0.55–0.76) ***^		
Optimism	-				-				0.67	(0.55–0.81) ***^		

^‡^ Age, sex, education level, employment and relationship status; ^†^ In receipt of following income support: single parent payment, unemployment allowance, youth allowance, disability support, carer payment; D = Tjur’s coefficient of discrimination; * *p* < 0.05; ** *p* < 0.01; *** *p* < 0.001; ^ Predictor made an independent significant contribution in the third and final model; adjusted odds ratios (aORs) reported are for the model in which the predictors were added.

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
