# Peer review of "Belonging and Inclusivity Make a Resilient Future for All: A Cross-Sectional Analysis of Post-Flood Social Capital in a Diverse Australian Rural Community"

_ijerph, 2020, doi:10.3390/ijerph17207676_

Round 1

Reviewer 1 Report

This manuscript presents the results of a cross-sectional survey of a diverse set of residents in a rural community that experienced extensive flooding in 2017. The study aimed to uncover associations between two distinct forms of social capital and psychological distress and PTSD across the community as a whole, as well as understanding differential impacts on marginalized groups, including Aboriginal and Torres Strait Islander residents and individuals receiving income support.

Both of these aims are important and the study’s overall approach to addressing them is robust, including engaging community members to increase participation, incorporating multiple measures via a confirmatory factor analyses for each form of social capital, and relying on a clinical screening tool to assess PTSD. Conversely, the study appears to be underpowered to explore relationships among Aboriginal and Torres Strait Islander survey respondents alone, and there are also a few other minor methodological issues that preclude publication as-is.

Specific comments on how to improve individual sections of the manuscript appear below.

Introduction
Lines 71-72: Please briefly expand upon the potential bidirectional relationship between social capital and mental health in order to provide a foundation to address the potential impacts of using a cross-sectional design among the strengths and limitations.

Materials and Methods
Lines 116-119: Please clarify how the choice between the short and long versions of the questionnaire was offered to participants (in initial discussions with study recruiters?).
Lines 141-143: Please provide additional justification for the selection of the Brief Weather Disaster Trauma Exposure and Impact Screen, including why it was chosen over more broadly used tools and the extent to which it has been previously validated, if at all.
Line 169: Trait optimism is not a standard component of individual social cohesion and it is being drawn from a tool designed to assess life orientation: please explain why this measure was selected for inclusion and cite prior research that has included it as a measure of cohesion.
Lines 203-206: Please delineate the fixed and random effects specified within the hierarchical regression model.

Results
As stated earlier, although no power calculation is provided to offer a statistical test of the necessary sample size, the extremely wide confidence intervals of the ORs reported for Aboriginal respondents alone (n = 67) is a clear indication that the study is underpowered for such an examination. The authors’ interest in examining this group is clear, and they make a strong argument for the contribution such an analysis would make to the literature, but the inclusion of these results along with those for groups with an adequate sample size undermines the manuscript as a whole. I strongly believe these analyses and the related content across the rest of the manuscript should be removed before publication.

Discussion
Lines 524-526: Due to the cross-sectional design of the study, the authors should be cautious about statements that may appear to assert causality, such as this sentence. Please check the manuscript carefully to ensure that the description of relationships fairly and fully reflects the selected study design.
Lines 571-572: Similarly, it might be helpful to integrate an additional sentence or two here discussing the ways in which the design limits exploration of the complex pathways linking the various factors under study, in addition to precluding a statement regarding the protective effects of social capital.
Lines 593-595: The authors do an excellent job of describing the ways in which community engagement increased the study’s robustness, particularly in terms of the sampling approach. Other roles played by the Community Advisory Groups cited in the Materials and Methods section are less clear, however, and the manuscript also omits any discussion of how such community engagement flowed through to the dissemination of results. Some additional discussion of these issues here – as well as in the Introduction, to the extent appropriate – would be beneficial.

Conclusions
Lines 617-620: These statements do not seem to align with the stated aims of the study or the bulk of the presentation of results and discussion, and the second sentence in particular does not reflect a finding by the authors themselves. I would recommend revising them to clarify the distinction between the study’s contributions and what is known about post-disaster response more broadly.

Tables
Although this may be due to the provisional format of the tables, they have a few minor issues that could be improved in advance of the final proof. These include the fact that some cells are centered while others are not and the lack of breaks between individual elements of the captions (which would increase readability).

Author Response

Response to Reviewer 1 Comments

Point 1: Introduction Lines 71-72: Please briefly expand upon the potential bidirectional relationship between social capital and mental health in order to provide a foundation to address the potential impacts of using a cross-sectional design among the strengths and limitations.

Response 1: As the reviewer suggests, we have expanded on the reciprocal relationship between mental health and social capital that has previously been demonstrated through longitudinal studies (lines 72-77).

Point 2: Materials and Methods Lines 116-119: Please clarify how the choice between the short and long versions of the questionnaire was offered to participants (in initial discussions with study recruiters?).

Response 2: The study’s self-completion survey tool was quite lengthy taking approximately 25 mins to complete. Hence, we provided participants an option of completing a shorter version. The instructions were included at the start of the survey, advising participants that the questionnaire had two parts along with the estimated amount of time it took to complete each (Part 1 - 15 mins and Part 2 - 10 mins). Instructions were again provided at the end of Part 1 to remind participants of their choice. A prize draw was offered as an incentive for respondents to complete the full questionnaire but was not promoted as part of recruitment for the study. This has been clarified in lines 131-138.

Point 3: Materials and Methods Lines 141-143: Please provide additional justification for the selection of the Brief Weather Disaster Trauma Exposure and Impact Screen, including why it was chosen over more broadly used tools and the extent to which it has been previously validated, if at all.

Response 3: The Brief Weather Disaster Trauma Exposure and Impact Screen contains four measures adapted from Australian research that includes a predictive screen index for PTSD and depression in adults following trauma (O’Donnell et al, 2008) and information about post-natural disaster PTSD in children and adolescents (McDermott et al, 2012). It has been used previously in a similar study examining mental health outcomes following extensive flooding in Brisbane, Queensland in a neighbouring area north of our study region (Clemens et al, 2013). We chose the yes/no item from the screen “Are you still currently distressed about what happened during the flood?” for the purpose of comparability and as it is directly linked to the flooding event, to partially address the issue of attribution. We have added further information into the manuscript (lines 175-183) and more information about the development of the measure in new Appendix A.

Point 4: Materials and Methods Line 169: Trait optimism is not a standard component of individual social cohesion and it is being drawn from a tool designed to assess life orientation: please explain why this measure was selected for inclusion and cite prior research that has included it as a measure of cohesion.

Response 4: Berry (2007) provides an overview of the link between optimism and life success and wellbeing in general including succeeding in employment, greater cognitive ability and less absenteeism at school, success in sports and politics and improved health overtime for men with HIV and older people. Dispositional optimism (a tendency to expect good rather than bad outcomes) has been tested previously as a component of personal social cohesion with the expectation that those with higher levels of trait optimism would contribute to and experience higher levels of social cohesion. We have referenced prior research where optimism was related to social trust and sense of belonging and greater optimism was independently related to less distress, including for Aboriginal Australians (Berry & Shipley, 2009; Berry, 2009). We have included this detail in the manuscript in lines 207-211.

Point 5: Materials and Methods Lines 203-206: Please delineate the fixed and random effects specified within the hierarchical regression model.

Response 5: We employed hierarchical logistic regression as a way to show if variable blocks (socio-demographic characteristics, flood exposure, community participation and personal social cohesion) explained a statistically significant amount of variance (Tjur’s coefficient) in mental health outcomes after accounting for other variables. Our hierarchical regression was a framework for model comparison rather than a multi-level mixed effects statistical analysis, therefore, fixed and random effects are not specified.

Point 6: Results - As stated earlier, although no power calculation is provided to offer a statistical test of the necessary sample size, the extremely wide confidence intervals of the ORs reported for Aboriginal respondents alone (n = 67) is a clear indication that the study is underpowered for such an examination. The authors’ interest in examining this group is clear, and they make a strong argument for the contribution such an analysis would make to the literature, but the inclusion of these results along with those for groups with an adequate sample size undermines the manuscript as a whole. I strongly believe these analyses and the related content across the rest of the manuscript should be removed before publication.

Response 6: We appreciate the reviewer’s point of view regarding the low sample size for Aboriginal respondents and we agree that a much larger sample is needed in future studies. We have highlighted the limitations of this (lines: 621-625). However, for the reasons outlined below, we strongly believe that removing this subgroup analysis will lead to Aboriginal and Torres Strait Islander people’s exclusion in local policy deliberations regarding disaster risk reduction thereby contributing to their ongoing marginalisation:

  • In the Strengths and Limitations section 4.5, we have acknowledged potential biases introduced by our sampling strategy and the low sample number for Aboriginal respondents and have been careful not to generalise results. As discussed, despite the limitations, our findings regarding Aboriginal social capital are in line with other previous studies using population representative samples and other study designs.
  • We have previously published multivariate analyses from our study using a similar sample number for Aboriginal respondents that indicated disproportionate impacts from the flood compared to non-Indigenous respondents (Matthews et al 2019; n=77)
  • In this paper, our aim was to examine relationships between social capital, flood exposure and psychological distress for the purpose of informing disaster risk reduction strategies. Where sample numbers were small, our analysis focused largely on direction of associations and whether they were consistent with our expectations and other research. Removing the analysis on Aboriginal respondents, potentially may mean benefits from this research will not flow equitably across community groups at a time when involvement of groups disproportionately affected is advocated for in disaster risk policy, such as the Sendai Framework for Disaster Risk Reduction (2015-2030) (United Nations Office for Disaster Risk Reduction, 2015) to which Australia is a signatory.
  • The draft manuscript has recently been approved for publication by the Aboriginal Health and Medical Research Council (AHMRC) Research Ethics Committee (a requirement of our ethics approval). If we were to remove the analysis, it would raise concerns within the committee regarding visibility and equity issues (manuscript approval letter attached).

Point 7: Discussion Lines 524-526: Due to the cross-sectional design of the study, the authors should be cautious about statements that may appear to assert causality, such as this sentence. Please check the manuscript carefully to ensure that the description of relationships fairly and fully reflects the selected study design. Lines 571-572: Similarly, it might be helpful to integrate an additional sentence or two here discussing the ways in which the design limits exploration of the complex pathways linking the various factors under study, in addition to precluding a statement regarding the protective effects of social capital.

Response 7: We agree with the reviewer and have modified the sentence within the discussion to remove inference of causality (see lines 567-570). We have also added a sentence (lines 614-616) about the limitations of cross-sectional design when untangling the complexity of pathways between weather-related disasters, social capital and mental health.

Point 8: Discussion Lines 593-595: The authors do an excellent job of describing the ways in which community engagement increased the study’s robustness, particularly in terms of the sampling approach. Other roles played by the Community Advisory Groups cited in the Materials and Methods section are less clear, however, and the manuscript also omits any discussion of how such community engagement flowed through to the dissemination of results. Some additional discussion of these issues here – as well as in the Introduction, to the extent appropriate – would be beneficial.

Response 8: We thank the reviewer for their positive comments regarding our participatory engagement approach. Further detail about this has been published in our study protocol which we have referenced in the Methods section. As the reviewer suggests, we have now provided a summary of our community engagement approach for the benefit of the reader in the Introduction (lines 82-92) and Strengths and Limitations section (lines 644-647).

Point 9: Conclusions Lines 617-620: These statements do not seem to align with the stated aims of the study or the bulk of the presentation of results and discussion, and the second sentence in particular does not reflect a finding by the authors themselves. I would recommend revising them to clarify the distinction between the study’s contributions and what is known about post-disaster response more broadly.

Response 9: We have modified the conclusion section so that it reflects the main results from the analysis.

Point 10: Tables - Although this may be due to the provisional format of the tables, they have a few minor issues that could be improved in advance of the final proof. These include the fact that some cells are centered while others are not and the lack of breaks between individual elements of the captions (which would increase readability).

Response 10: Initial formatting by the journal has improved consistency of table presentations. We will work with the journal during proofing stage to ensure table readability.

References

Berry, H.; George, E.; Butterworth, P.; Rodgers, B.; Caldwell, T. Intergenerational Reliance on Income Support: Psychosocial Factors and Their Measurement Australian Government: Department of Families, Housing, Community Services and Indigenous Affairs: Canberra, Australia, 2007, doi:10.2139/ssrn.1728550.

Berry, H.L.; Shipley, M. Longing to belong: personal social capital and psychological distress in an Australian coastal region; Australian Government: Department of Families, Housing, Community Services and Indigenous Affairs: 2009, ISBN 978‑1‑921647‑08‑6.

Berry, H.L. Social capital and mental health among Aboriginal Australians, New Australians and Other Australians living in a coastal region. Aust J Adv Mental Health 2009, 8, 142-154, doi:10.5172/jamh.8.2.142.

Clemens, S.; Berry, H.; McDermott, B.; Harper, C. Summer of sorrow: measuring exposure to and impacts of trauma after Queensland’s natural disasters of 2010–2011. Med J Aust 2013, 199, 552-555, doi:10.5694/mja13.10307.

Matthews, V.; Longman, J.; Berry, H.L.; Passey, M.; Bennett-Levy, J.; Morgan, G.G.; Pit, S.; Rolfe, M.; Bailie, R.S. Differential Mental Health Impact Six Months After Extensive River Flooding in Rural Australia: A Cross-Sectional Analysis Through an Equity Lens. Front Public Health 2019, 7, doi:10.3389/fpubh.2019.00367.

McDermott, B.; Berry, H.; Cobham, V. Social connectedness: A potential aetiological factor in the development of child post-traumatic stress disorder. Aust N Z J Psychiatry 2012, 46, 109-117, doi:10.1177/0004867411433950.

O'Donnell, M.; Creamer, M.; Parslow, R.; Elliott, P.; Holmes, A.; Ellen, S.; Judson, R.; McFarlane, A.; Silove, D.; Bryant, R. A Predictive Screening Index for Posttraumatic Stress Disorder and Depression Following Traumatic Injury. J Consult Clin Psychol 2008, 76, 923-932, doi:10.1037/a0012918.

United Nations Office for Disaster Risk Reduction. Sendai Framework for Disaster Risk Reduction 2015-2030. Available online at: https://www.undrr.org/publication/sendai-framework-disaster-risk-reduction-2015-2030.

Reviewer 2 Report

I am a pleasure to review your manuscript. I think this is a significant topic. This article showed that informal social connectedness and belonging were important factors, and associated with reduced risk of psychological distress among the aboriginal respondents/ respondents in financial hardship following the flood disaster. The author(s) did excellent presenting the contents. I would ask the author(s) to make modifications to your manuscript.

[Overall points]
In your fondings, the prevalence of traumatic reactions (probably PTSD) following the severe flood was higher among aboriginal respondents and respondents in financial hardship. What factors/experiences are related to high PTSD symptoms among the aboriginal respondents/ respondents in financial hardship? Why the prevalence of PTSD symptoms are higher than those who are the general community respondents?

Could you please mention ethical considerations in this study because there was no description of ethical consideration. If it has been approved by any ethical committee, please add the number and date of approval.

[Specific points]
<Abstract>: If possible, it is better to describe the statistical values (e.g. adjusted odds ratio) in the abstract.

<Introduction>
L73: Please show the impact or damages on the disaster-stricken area of the 2017 flood disaster(e.g. Number of casualty or missing people, the number of damaged houses).

L105: This study aims to use these findings to highlight what might or might not work in intervention design to assist community groups to strengthen social capital and adaptive capacity within this flood-prone region. However, it was difficult to understand what results were expected from the survey. So, it is necessary to mention the hypothesis in this study.

<Materal and Method>
L120: You had better explain detailed sampling methods while using a figure.

L137: It needs to explain what CEI stands for. (Cumulative Exposure Index (CEI))

L223: Could you mention about the psychological distress scale? (Please show the detail information about the Brief Weather Disaster Trauma Exposure and Impact Screen, cut-off points, and so on)

<Results>
L245: It is necessary to explain the full spelling of CFI, RMSEA and (95%)CI in the footnote of Table2.

L259: The footnote of table 3 is quite long to read. It may be better to move to the main text.

L307: Table 6 and 7 are confusing to see. It needs a more organized arrangement on these tables.

L323: I think it needs to explain the results according to table 6 (ongoing distress) and table 7 (PTSD) not based on the aboriginal respondents/ respondents in financial hardship/ general community respondents.

<Discussion>
L570: The case of aboriginal respondents is 67, which number maybe not enough to assess using multivariate analysis. I think it is a severe limitation in this study.

L570: Also, since there was no measurement about changes in social capital between pre- and post-flood disaster, it was difficult to assess whether your findings come from the specific situation following the flood disaster or not. I think it is a limitation in this study. (e.g. It is able to continue interacting with acquaintances before the disaster.)

Author Response

Response to Reviewer 2 Comments

Point 1: In your findings, the prevalence of traumatic reactions (probably PTSD) following the severe flood was higher among aboriginal respondents and respondents in financial hardship. What factors/experiences are related to high PTSD symptoms among the aboriginal respondents/ respondents in financial hardship? Why the prevalence of PTSD symptoms are higher than those who are the general community respondents? 

Response 1: In previous analyses from our study published elsewhere, rates of probable PTSD across all respondents were particularly elevated in those who had their home inundated, experienced lengthy displacement and faced multiple exposures. Using geographical information systems methods, we have mapped the 2017 flood-affected areas in the region and combined this with Australian national census data which showed that residents in the flood footprint had higher levels of socio-economic vulnerability (Rolfe et al, 2020). As stated in the Abstract (lines 15-17) and Introduction (lines 111-113), Aboriginal respondents and respondents on income support fall into this high-risk group – they were more likely to have their homes inundated and to be displaced from their homes for six months or more (Matthews et al, 2019). It is also recognised that there is likely to be pre-existing psychological morbidity for these groups due to their poorer underlying socio-economic status, hence we cannot be sure whether flood experiences directly caused these outcomes. We have added a sentence about this in the Strengths and Limitations section (lines 618-620). To overcome these limitations, we included PTSD measure specifically related to the flood and we adjusted our analyses by a wide variety of socio-demographic factors known to predict psychological morbidity.

Point 2: Could you please mention ethical considerations in this study because there was no description of ethical consideration. If it has been approved by any ethical committee, please add the number and date of approval.

Response 2: Thank you for pointing out this oversight. The study ethics approval has been added to the Materials and Methods section lines 159-162.

Point 3: Abstract - If possible, it is better to describe the statistical values (e.g. adjusted odds ratio) in the abstract.

Response 3: Given the Abstract word limit (200 words) and the number of adjusted odds ratio results that could be reported for informal social connectedness and belonging related to ongoing distress and PTSD across the three respondent groups, we thought it preferable to describe direction of associations in the summary rather than magnitude. However, if the journal feels strongly about AOR inclusion in the Abstract, we can amend.

Point 4: Introduction L73 Please show the impact or damages on the disaster-stricken area of the 2017 flood disaster(e.g. Number of casualty or missing people, the number of damaged houses).

Response 4: There is limited availability from authoritative sources of infrastructure damage and casualty statistics within our region due to the 2017 flood. Media reports suggest up to 6 deaths within New South Wales. Within the main population centre, Lismore, where the levee was overtopped for the first time, hundreds of residential and commercial businesses were flooded and several thousand residents and business operators were evacuated. We have drawn on an independent report into the State Emergency Response that documented the severity of the flood in the region to provide scale of the disaster within Australia (line 79).

Point 5: Introduction L105 This study aims to use these findings to highlight what might or might not work in intervention design to assist community groups to strengthen social capital and adaptive capacity within this flood-prone region. However, it was difficult to understand what results were expected from the survey. So, it is necessary to mention the hypothesis in this study.

Response 5: In the Introduction, we have elaborated on our ‘flood impact framework’ which sets out theorised pathways between flood exposure and mental health and the potential mediating factors in between (lines 88-92). Further information is available from our study protocol which we have referenced (Longman et al, 2019). Our hypothesis for this paper is outlined in lines 93-95: greater levels of community participation and social cohesion would reduce the likelihood of ongoing distress and probable PTSD and this would vary for different socio-economic groups.

Point 6: Materials and Methods L120 You had better explain detailed sampling methods while using a figure.

Response 6: We have further elaborated on the purposive snowball sampling technique aimed at recruiting a broad cross-section of the community including hard-to-reach groups such as people in financial hardship (lines 149-155).

Point 7: Materials and Methods L137 It needs to explain what CEI stands for. (Cumulative Exposure Index (CEI))

Response 7: The full title and derivation of the CEI (Cumulative Exposure Index) has been provided in lines 166-169.

Point 8: Materials and Methods L223 Could you mention about the psychological distress scale? (Please show the detail information about the Brief Weather Disaster Trauma Exposure and Impact Screen, cut-off points, and so on)

Response 8: In response to a similar comment from Reviewer 1 (see point 3), we have added new Appendix A to explain the development and source of the Brief Weather Disaster Trauma Exposure and Impact Screen. It is a dichotomous yes/no item that provides insight into whether ongoing stress and anxiety is directly related to the traumatic event.

Point 9: Results L245 It is necessary to explain the full spelling of CFI, RMSEA and (95%)CI in the footnote of Table2.

Response 9: As suggested, we have added explanation of these acronyms into footnote of Table 2.

Point 10: Results L259 The footnote of table 3 is quite long to read. It may be better to move to the main text.

Response 10: To improve readability, we have taken out the text within the footnote that already appears within the main text.

Point 11: Results L307 Table 6 and 7 are confusing to see. It needs a more organized arrangement on these tables.

Response 11: Reviewer 1 has also made similar comments about the table formatting (see point 10). We will work with the journal during proofing stage to ensure readability.

Point 12: Results L323 I think it needs to explain the results according to table 6 (ongoing distress) and table 7 (PTSD) not based on the aboriginal respondents/ respondents in financial hardship/ general community respondents.

Response 12: We acknowledge the results could also be structured according to mental health outcome (we did this originally), however, as the focus of the paper is about how social capital has different effects on mental health for marginalised groups relative to other participants, we felt it best to structure the narrative (in both results and discussion) according to respondent group. We felt this improved the flow of the paper.

Point 13: Discussion L570 The case of aboriginal respondents is 67, which number maybe not enough to assess using multivariate analysis. I think it is a severe limitation in this study. Also, since there was no measurement about changes in social capital between pre- and post-flood disaster, it was difficult to assess whether your findings come from the specific situation following the flood disaster or not. I think it is a limitation in this study. (e.g. It is able to continue interacting with acquaintances before the disaster.)

Response 13: While we recognise the low sample number is a limitation and have discussed this within the Strengths and Limitations section (lines 621-623), we strongly believe that the Aboriginal subgroup analysis should be retained for reasons highlighted above under reviewer 1’s point 6. A copy of the manuscript approval from the Aboriginal Health & Medical Research Council Ethics Committee is attached as support.

References

Longman, J.M.; Bennett-Levy, J.; Matthews, V.; Berry, H.L.; Passey, M.E.; Rolfe, M.; Morgan, G.G.; Braddon, M.; Bailie, R. Rationale and methods for a cross-sectional study of mental health and wellbeing following river flooding in rural Australia, using a community-academic partnership approach. BMC Public Health 2019, 19, 1255, doi:10.1186/s12889-019-7501-y.

Matthews, V.; Longman, J.; Berry, H.L.; Passey, M.; Bennett-Levy, J.; Morgan, G.G.; Pit, S.; Rolfe, M.; Bailie, R.S. Differential Mental Health Impact Six Months After Extensive River Flooding in Rural Australia: A Cross-Sectional Analysis Through an Equity Lens. Front Public Health 2019, 7, doi:10.3389/fpubh.2019.00367.

Rolfe, M.I.; Pit, S.W.; McKenzie, J.W.; Longman, J.; Matthews, V.; Bailie, R.; Morgan, G.G. Social vulnerability in a high-risk flood-affected rural region of NSW, Australia. Nat Hazards 2020, 101, 631-650, doi:10.1007/s11069-020-03887-z.

Round 2

Reviewer 1 Report

Thank you for your close attention to the feedback provided during the initial review. I believe the revisions you have made to the original manuscript as a result greatly improve it and adequately address the issues raised.

Although I still feel that the Aboriginal-only subanalyses are limited by their sample size, you have persuasively argued that these subanalyses as essential due to ethical and representative concerns. Integrating some of this justification into the manuscript itself would be beneficial, but I do not consider it a mandatory requirement for publication.

Author Response

Response to Reviewer 1 Comments

Point 1: Thank you for your close attention to the feedback provided during the initial review. I believe the revisions you have made to the original manuscript as a result greatly improve it and adequately address the issues raised.

Although I still feel that the Aboriginal-only subanalyses are limited by their sample size, you have persuasively argued that these subanalyses as essential due to ethical and representative concerns. Integrating some of this justification into the manuscript itself would be beneficial, but I do not consider it a mandatory requirement for publication

Response 1: We thank the reviewer for their suggestions on improving the manuscript and their positive comments on the initial revisions made. In response to this latest suggestion, we have added justification into the Limitations section regarding our decision to include sub-analyses of Aboriginal respondents (lines 616-621): i.e., focusing on direction of associations, whether they were consistent with our expectations and other research and the value these results bring in informing future work on disaster risk reduction strategies for this population group.